# Sensing and Artificial Intelligent Maternal-Infant Health Care Systems: A Review

**DOI:** 10.3390/s22124362

**Published:** 2022-06-09

**Authors:** Saima Gulzar Ahmad, Tassawar Iqbal, Anam Javaid, Ehsan Ullah Munir, Nasira Kirn, Sana Ullah Jan, Naeem Ramzan

**Affiliations:** 1Department of Computer Science, Wah Campus, COMSATS University Islamabad, Islamabad 45040, Pakistan; saimagulzarahmad@ciitwah.edu.pk (S.G.A.); tassawar@ciitwah.edu.pk (T.I.); anam.javaid@ciitwah.edu.pk (A.J.); 2School of Computing, Engineering and Physical Sciences, University of the West of Scotland, Glasgow G72 0LH, UK; nasira.kirn@uws.ac.uk; 3School of Computing, Edinburgh Napier University, Edinburgh EH10 5DT, UK; s.jan@napier.ac.uk (S.U.J.); naeem.ramzan@uws.ac.uk (N.R.)

**Keywords:** healthcare, maternal, infant, artificial intelligence, machine learning, wearable sensors, wireless sensors

## Abstract

Currently, information and communication technology (ICT) allows health institutions to reach disadvantaged groups in rural areas using sensing and artificial intelligence (AI) technologies. Applications of these technologies are even more essential for maternal and infant health, since maternal and infant health is vital for a healthy society. Over the last few years, researchers have delved into sensing and artificially intelligent healthcare systems for maternal and infant health. Sensors are exploited to gauge health parameters, and machine learning techniques are investigated to predict the health conditions of patients to assist medical practitioners. Since these healthcare systems deal with large amounts of data, significant development is also noted in the computing platforms. The relevant literature reports the potential impact of ICT-enabled systems for improving maternal and infant health. This article reviews wearable sensors and AI algorithms based on existing systems designed to predict the risk factors during and after pregnancy for both mothers and infants. This review covers sensors and AI algorithms used in these systems and analyzes each approach with its features, outcomes, and novel aspects in chronological order. It also includes discussion on datasets used and extends challenges as well as future work directions for researchers.

## 1. Introduction

Maternal and infant health is indispensable for a healthy society. The major issues that cause maternal health problems and even deaths include ectopic pregnancy [1], miscarriage [2], high blood pressure that leads to preeclampsia [3,4], and failure in progress of labor that might lead to Cesarean section (C-section) [5]. In addition, iron deficiency is another vital factor of prenatal and postnatal health complications that can be caused due to antepartum or post-partum hemorrhage [6], retained placenta [7], vaginal infection during delivery [8,9], and many other similar problems. In the case of neonatal or newly born babies, a few of the major issues that may cause health problems include growth retardation in uterus, birth asphyxia during delivery, infection due to maternal vaginal delivery at the time of birth, shoulder dystocia, meconium aspiration, septicemia, and premature delivery/premature lungs or any other congenital abnormalities. The complications of both maternal and infant health need to be identified and communicated in a timely manner to health caretakers and professionals take action against them. However, approaching medical assistance in time is a difficult job, especially in rural areas. In the current era, information and communication technology has enabled the medical field to address complications of maternal and infant health in a timely manner.

Development in technology empowers the health sector to offer improvised services for maternal and infant health using sensing, AI, and computing platforms at the doorstep. These strands are depicted in a general taxonomy as shown in Figure 1. The sensor-enabled systems detect biological information, AI approaches exploit this biological information to predict the health condition, and computing platforms offer optimized communication, huge storage and computing capabilities for exponentially increasing data with the increasing Internet of Things (IoTs).

Currently, sensing technology has improved patient pregnancy care, infant healthcare, patient communication, and real-time monitoring of several diseases using sensors, wearable gadgets and devices [10]. The relevant literature reports the use of a variety of solutions, such as in ectopic pregnancy, that can be detected early through scans and blood tests by using high resolution and portable ultrasound [11] and complete blood count (CBC) blood test machines [12]. In the case of a miscarriage scans, timely dilation and curettage (D&C) and hemoglobin (Hb) level monitoring can help to minimize the relevant complications. Similarly, for blood pressure (BP) [13] monitoring, a variety of digital devices have been designed. Cardiotocography (CTG)/tracing of fetal heart [14] and anomaly scans can reduce fetal deaths and stillbirths and can help in the early detection of congenital abnormalities, pregnancy care [15], hypertension monitoring [16], diabetes [1,17,18,19], mother and fetal care [20], detection of postpartum depression [21], and similar pregnancy complications. Consequently, wearable sensor advances can improve patient–supplier connections for successful pregnancy wellbeing. In addition, for infant healthcare, technology-enabled, well-equipped and intelligent incubators and nasogastric (NG) tubes for nasogastric feed premature infants can improve in time response [22,23].

Similarly, machine learning (ML) approaches may help to predict abnormal behavior in mother and infant health. The immense growth of ML algorithms to monitor mother and infant health in the earlier stages of pregnancy may help doctors to tackle complications. In the present era, ML approaches are being used for the prediction of preterm birth risk [24], detection of wild stress [25], prenatal risk [26], postpartum depression [27], and congenital heart disease [28] among pregnant women. In addition, the ML approaches have potential to predict the infant’s health status and to monitor the brain and general growth of baby. A variety of algorithms such as Support Vector Machine (SVM) [29], Artificial Neural Networks (ANN) [30], Regression Analysis (RA), and Random Forest are among the popular methods that are being applied to determine the best pregnancy outcomes [31]. Therefore, with this recently discovered interest in the possibilities of ML in medical services and specifically for maternal and infant health, it has been expressed to be the foundation in maternal and infant care transmission [32].

The sensing and artificially intelligent healthcare systems must store and compute huge volumes of data. To address the issues of big data and computing capabilities, different computing platforms such as edge [33,34,35,36], fog [37,38,39,40,41,42], and cloud [43,44,45,46] provide great support to store, process and classify data. A general architecture exploiting these technologies for maternal and infant healthcare systems is depicted in Figure 2. First, the data are collected from the sensors that monitor the patients’ condition by measurable health parameters. Second, the fog layer plays a mediating role between the edge/IoT devices and the cloud to enhance computing efficiency and response time. The fog layer uses protocols to generate effective outcomes. Moreover, the data are analyzed, processed, and classified using classification algorithms for decision making at the fog layer. Later, the data are stored in the cloud layer for further processing. In the cloud layer, messages and alerts are generated to the doctors, health workers, and family in case of emergency to take immediate action [47].

Wearable sensor-based technologies and ML approaches have the potential to improve patient provider interactions for effective pregnancy health management, thus replacing the traditional healthcare settings with a high positive rate [10,48,49]. This study reviews existing wearable sensing technologies and ML techniques used for better diagnosis of various pregnancy complications and infant health monitoring. In this survey, our research contributions can be summarized as follows:Classification of the healthcare systems based on sensors used and ML techniques exploited.Each healthcare system, framework or model are reviewed in chronological order.Discourse on the real-world datasets and their details in terms of attributes, size, format, etc.Comprehensive discussion about current research issues and challenges linked with these systems.Explore potential future work directions for researchers in the relevant domain.

In Section 2, logical rationale of the review article is presented to set the motivation of this work and its differences from existing review articles. Section 3 discusses the materials and methods followed for the literature review. In Section 4, wearable sensing technologies used and AI/ML techniques exploited in both maternal and infant healthcare systems are discussed, which will enable a reader to seek knowledge of advancement and limitations in sensing and AI/ML technologies used in both maternal and infant healthcare systems. In Section 5, datasets used in sensing and AI/ML-based maternal and infant healthcare systems are presented. Section 6 sums up the reviewed articles in context of challenges and future opportunities for work. The last section of conclusions presents an overview of the paper for quick readers.

## 2. Rationale

This review is motivated by a need to understand the sensing devices and AI/ML techniques in both maternal and infant healthcare systems. The existing literature suggests that current reviews on sensors and AI/ML techniques in maternal and infant healthcare systems are limited in scope. This article presents sensing devices and AI/ML techniques in both maternal and infant health monitoring systems in a single review. A comparison with existing relevant review papers is presented in Table 1, which clearly depicts the major differences and novelties of this review article. By this review, we will discover the wearable sensing technologies and AI/ML techniques exploited in both maternal and infant healthcare systems that will lead to challenges and opportunities for future works in this domain.

## 3. Methods

The literature review considers the accessible scholarly articles in the English language from four databases, namely, PubMed, Semantic Scholar, National Center of Biotechnology Information (NCBI) and some other databases for the last three years. These databases are selected because they have the most relevant and reputed peer-reviewed list of journals where research relevant to sensing and AI/ML-based healthcare systems is usually published. These scholarly databases are selected since these are mostly considered by existing review articles. The articles were initially searched for based on keywords such as healthcare, maternal, infant, artificial intelligence, machine learning, wearable sensors, wireless sensors, and remote health monitoring. These keywords were searched for in the title, abstract, and article keywords sections, and the search returned 241 articles. The duplicates were eliminated, and a total of 173 articles were left. In the next step, the number of relevant articles was further narrowed down based on the examination of title and abstracts, which lowered the number to 96 articles. A pictorial view of the material and methods followed for the review is shown in Figure 3.

## 4. Sensing and AI Based Classification of Health Monitoring Systems

Maternal and infant monitoring systems continuously monitor the condition of mother and fetus throughout pregnancy and help them obtain treatment on time in the case of risk factors such as blood pressure, hypertension disorders, diabetes, anemia, and other related issues. Similarly, the health monitoring systems for infants monitor the condition and risk factors that include fetal growth, congenital abnormalities, heart rate, glucose level and other related issues for the baby after birth.

This section discusses both maternal and infant health monitoring systems in a specific order, as shown in Figure 4. First, maternal healthcare systems are discussed in the context of sensors used and AI/ML approaches exploited in Section 4.1. and Section 4.2, respectively. Second, infant healthcare systems are reviewed with respect to sensing technologies considered and AI/ML techniques used in Section 4.3. and Section 4.4 respectively. Both healthcare systems are discussed in chronological order, and the discussion focuses on the crucial achievements and sensing/AI technologies exploited in these systems.

### 4.1. Sensor Used in Maternal Healthcare System

The use of wearable and wireless sensors for the healthcare of pregnant women reported a great improvement in women’s health as well as in solving pregnancy complications. These wearable wireless sensors help to provide knowledge, to detect breast cancer chronic diseases, and to motivate women to control their weight, diabetes, and mental health [52,53]. In the health status of pregnancy, the use of wearable wireless sensors provides remote monitoring to pregnant women [10].

The advanced development of different eHealth applications and the use of wearable and wireless sensors explore the new opportunities to provide better clinical monitoring and diagnosis to pregnant women as compared to conventional healthcare techniques. The use of advanced sensor technology in the field of maternal health helps to detect the complications of pregnancy in the early stage and to motivate and provide better health facilities to improve these complications. Currently, these wireless or wearable sensors are used to detect risk factors such as blood pressure, heart rate, breathing rate, etc., comprising the patient’s lifestyle and behavior in the subclinical period of an unfavorable pregnancy result [54].

Hypertension or preeclampsia is one of the major complications in pregnancy. The outcomes of this can be a danger to the life of the mother or child, including pre-birth or low birth weight. A wearable technology model is proposed [55] that can be used to control and monitor preeclampsia in pregnant women and can perform real-time data analysis. The prior models are improved by including a chain of new healthcare parameters. These healthcare parameters can put a stop to preeclampsia issues in pregnancy. This study was carried out in a health center in Lima, Peru. In the study, “VO7” wearable devices that can measure blood pressure, heart rate, and steps were distributed among different pregnant women along with usage instructions. The device transfers all the recorded data of about 30 min of the patients to a mobile application, which further stores data in the Systems, Applications and Products High-Performance Analytic Appliance Application (SAP HANA) database. The wearable device needs to be connected to the mobile application via Bluetooth, and it needs to be connected to the internet. The results are presented in graphs that show the patients’ heart rate and blood pressure. The results can further help health center professionals to monitor them. According to the reported results, 7% of maternal deaths were reduced, and 11% of patients were controlled by using the proposed method. The pregnant women felt confident, as they could monitor themselves, and they expressed their satisfaction with the model. Due to the better graphics, friendly graphical user interface (GUI), and alerts, the model is easy to use and non-invasive. The health center assured the model’s remarkable impact on their patients to prevent them from developing hypertension or preeclampsia.

One of the most significant preeclampsia complications is eclampsia, which is quite rare but which is a severe issue. Eclampsia causes seizures during pregnancy, which happen as a result of high blood pressure. For monitoring eclamptic pregnant patients to avoid severe pregnancy problems, a method is proposed in [56] that exploits wireless 5G sensors. In the study, an indoor environment experiment was performed using a wireless transceiver to watch an eclamptic seizure patient’s body motion. The experiment provided unique wireless data, which resulted in fetch information on the wireless channel. The recorded data were divided into four different body motions: sitting, walking, lying on a bed, and seizures. Four classifiers, KNN, SVM, K-mean, and RF, were used during the experiment to classify the data. The results for each classifier were compared to check the performance of the classification algorithms. These results were recorded using the six-performance metrics, including accuracy, specificity, f-measures, recall, kappa, and precision. According to the results, SVM outperformed other classifiers in term of accuracy, with a 0.97 kappa coefficient value and low error rate. The proposed method achieved promising results in the detection of seizures through body movements. Future challenges can be to design a process that can predict the seizure disease in a timely manner to prevent severe injury to the pregnant patient.

Gestational diabetes mellitus (GDM) is a severe threat to the mother’s and child’s health, which is often neglected. Patients with GDM experience several pregnancy complications, including high blood pressure, obstructed labor, and considerable birth weight. The existence of two proteins including hemoglobin (Hb) and glycated hemoglobin (HbA1c) helps to determine GDM. For the determination of Hb and HbA1c, a model is proposed that uses flexible electrochemical sensors comprising double imprinted nanocubes [57]. In the study, blood samples of diabetic and healthy pregnant women were examined. The performed experiments showed that the proposed sensor has the potential for a simultaneous redox reaction of electrochemical catalyzation for both Hb and HbA1c. Moreover, the response generated remained unchanged after the 450 bends. The reported results show that the sensor has potential for a wide range of applications in diabetes mellitus monitoring.

Due to physician shortages in geographical rural areas, access to healthcare during pregnancy can be challenging for pregnant women. Multiple numbers of pregnant women visiting for healthcare checkups can also consume time and burden a doctor. Currently, cardiotocography is used for fetal heart rate (FHR) and maternal heart rate (MHR) monitoring as a standard healthcare tool for pregnant women. All medical and healthcare professionals need to adopt this cardiotocography technique setting. For cardiotocography, a wireless electrical and acoustic sensor wearable belt system known as the Invu System is proposed to compare and monitor FHR and MHR [58]. The Invu system was evaluated through a study in which 147 women with singleton pregnancies of more than 32 weeks of gestation, aged between 18 to 50 years, participated. The tests of the Invu System and cardiotocography were performed simultaneously for about equal to or more than 30 min, the passive eight electrical sensors and four acoustic sensors embedded on the belt gather the data. The acquired data are then digitized and shared wirelessly to an algorithm in a cloud server for analysis purposes. An algorithm removes the noisy data by preprocessing and encounters the independent heartbeats from electrical and acoustic sensors in the cloud server. The resultant detected heartbeats are then fused to calculate FHR and MHR. According to the reported results, a high significance correlation between the Invu System and cardiotocography for FHR (r = 0.92; *p* < 0.0001) and MHR (r = 0.97; *p* < 0.0001) has been recorded. The output of FHR and MHR with the Invu System shows that the results are similar to the current care standards. The Invu System is a passive technology that ensures safe and convenient patient monitoring, remotely or in a clinic,; thus, no adverse situations were reported during this study.

The presence of normoglycemia (NGA) and GDM in pregnant women makes it hard for them to fight against different severe issues. These issues may cause danger to the life of both the mother and the infant. Due to the burden of NGA and GDM in the South Asian region, a study was carried out [19] to detect NGA and GDM at the initial pregnancy level. For this study, a continuous glucose monitoring system (CGMS) is used that extracts and evaluates several glucose-related parameters in women with NGA and GDM. Women having pregnancies between 8 to 20 weeks were considered for this study. The study aimed to perform a comparison among those who have NGA and GDM. These eligible pregnant women then followed CGMS treatment. Out of 96 pregnant women, 58 were with NGA, and 38 were with GDM. The study results show that women with GDM have high peak glucose values (64.3 ± 11.6) versus women with NGA (60.0 ± 12.3). The time spent in this study with the NGA women group was slightly higher, at 98.2% versus the GDM women at 92.1%. This study observed that the comparative data from CGMS show glycemic pattern differences in women with NGA and women with GDM during early pregnancy. The evaluation of perinatal outcomes in pregnant women may be a future challenge that can reveal further confusion in such conditions.

In another study, to examine fetal hemodynamics and to evaluate health factors during pregnancy, intelligent ultrasound sensors (IUSS) were used in the proposed system [59]. For evaluation of the system, an experiment was performed where 237 pregnant women were selected and divided into three different groups. Group IA had 93 cases of gestational diabetes women, Group IB had 19 instances of pre-pregnancy diabetes women, and Group II had 125 standard control women. During the experiment, it was observed that the women’s blood glucose content in group IA and IB was relatively more significant than in group II women after a glucose tolerance test and fasting. According to the results, group IB women’s blood glucose content was higher than that of group IA women. Group IA and IB hemodynamic parameters were recorded differently from that of the standard control group II. Thus, using intelligent ultrasound sensors in the later pregnancy period to measure hemodynamics can predict pregnancy outcomes. Patient’s health status with abnormal glucose metabolism can be evaluated in the last pregnancy period using intelligent ultrasound sensors.

For the last few years, advancements in IoT technology have helped a lot in the health industry, especially for pregnant women. Exploiting the IoT technology, a system having an IoT platform, wearable devices, and cloud computing for smart maternal healthcare was proposed [46]. This platform makes it easy for pregnant women to improve their treatment quality and facilitate them to have a checkup with their doctors. It also reduces the workload for healthcare professionals and staff. The proposed IoT platform consists of three layers. The first layer is the perception layer that authenticates the user, takes intelligent control and gathers physiological data. These data are later transferred to the second layer, which is the network layer that performs routing of the data, transmits the data to cloud computing, and senses if there are any extended devices. The third layer is the application layer, which performs health management functions, detects diseases and monitors fetal health. The results were calculated over the data gathered from the questionnaire. SPSS statistical software was used for the analysis task. Almost 29.84% and 65.08% of pregnant women agreed and showed faith in the increased use of IoT technology and wearable devices in the medical healthcare field.

The death of babies during or before delivery is one of the major issues currently across the world. To check whether a baby is alive, multiple works on fetal movement have been recorded until now. Fetal movement is essential to monitor fetal growth, umbilical cord complications, gestational age, etc. Counting fetal movement daily can help health professionals to examine child health and pregnancy difficulties. For estimating fetal movement, a system based on optical fiber sensors was proposed in [60]. The data for this study were collected through strain management with the help of sensors. Independent component analysis (ICS) was applied to post-process the collected data. Fetal movement instances were observed by using high filtering. The collected information shows that the proposed prototype’s fetal movements are sensitive, much higher, and better against a mother’s perception. There are many devices for measuring fetal movement, but optical fiber sensors proved to be beneficial. Multiplex capability, minimal size, and flexibility are few important advantages of using optical fiber sensors for fetal movement counting. For the future, a wearable belt can be designed and developed with optical fiber sensors that can reduce motion artifacts, classify moment type, and use ultrasound imaging validated with the proposed prototype. The above-discussed sensor-based maternal healthcare systems are summarized in Table 2.

### 4.2. AI/ML Techniques Used in Maternal Healthcare Systems

AI and ML methodologies including modern deep learning methods are helpful in detecting pregnancy outcomes. Accurate prediction methods and diagnosis during prenatal healthcare can help detect problems as early as possible. According to the relevant literature, supervised learning methods are more popular as compared to unsupervised learning methods [31]. Moreover, use of state-of-the-art ML methods to predict stillbirth, late stillbirth and preterm birth pregnancies is common [61]. It is reported that postpartum depression can often be severe, and its early prediction can help with preventive intervention or timely psychiatric admission of the new mother. In this regard, researchers developed a prediction model for women at risk of psychiatric admission [62]. Maternal data including delivery and neonatal data were used as predictors. In addition, many efforts aim at evaluating adverse pregnancy outcomes using ML. Different studies carried out to build the environment factors have immense impact on pregnancy outcomes. One of the studies was conducted using the electronic health data (EHR) of pregnant women who had live delivery at an urban medical center during 2015 to 2017. The reported results show that women living in high-quality built environments experience a different pattern of clinical events as compared to women in low-quality built environments. The reason behind such reported results was multi-purpose, and walkable communities have a low risk of postpartum depression (PPD) in urban settings [63]. ML techniques are being used in predicting the weight of new born babies based on features of the mother [64]. Although, ML has made a lot of progress in maternal healthcare, there are still many areas in maternal healthcare where advancements can be made using ML techniques. Figure 5 shows the general methodology to predict patient health status using ML algorithms, whereas all models exist for prediction and are discussed in this section, as shown in Table 3.

In order to predict the route of delivery, a supervised artificial neural network (ANN) [65] has been developed. The variables used for the input of ANN are gestational age at birth, maternal age, risk factors or disorders of the mother. The algorithm gives output in two forms including Cesarean (CS) or vaginal delivery (VI). For the experiment, data were obtained from the patients admitted for delivery, which were randomly selected to train the algorithm. The reported results show that the ANN-based system gives an efficiency rate of 97% and performs best in other statistical measures.

In another study, a prediction system was proposed to predict the congenital anomalies in the fetus [66]. Different binary classification models including SVM, decision forest, neural network, decision jungle and others were used to train and process the data to predict the fetal anomaly status. The data were obtained through a questionnaire from RadyoEmar radio diagnostics center in Istanbul, Turkey, and an e-health application was designed to obtain the parameters as an input. Experiments were performed, and the results show that when the Decision Forest algorithm is used to train the data, it gives the best predictions in terms of accuracy, AUC and F1 score.

Similarly, to monitor the chromosomal abnormalities and the risk of fetal trisomy, an ANN diagnostic system was proposed [67]. The system uses ANN to train the attributes of the data. The two-stage approach, which consists of diagnosis of aneuploidy cases and stratification of risk, was used. First, a blind set of pregnancies was classified into risk or no risk, using four markers. Then, the data with risky pregnancies were sent for further examination. Second, the risky pregnancies were further classified into three types, including high risk, no risk or moderate risk, using seven markers. The reported results show that the proposed approach outperformed the system using a mixture model as a classifier. In addition, the system has potential to be used in real-world applications in the medical field.

In another study, a deep-learning-based method was proposed [68] to accelerate the performance of subject-to-template 3D rigid registration and intersubjects and to estimate the capture range. The model was trained to find the volume of medical images and 3D positions of arbitrarily oriented subjects. In the proposed system, a regression-based CNN method was exploited to predict the 3D rotations and translations using image features. Finally, the models were trained using MRI scans and 3D pose of the fetal brain. The results of the study showed that the trained models achieved a 3D pose estimation with a wide capture range in real time in terms of mean squared error.

Ectopic pregnancy is one of the major causes of mortality and morbidity. To avoid complications, early diagnosis and the choice of treatment for such patients is decisive. In this regard, an AI algorithm-based clinical decision support system was developed [1]. The system uses two architectures: (1) consider the classifiers individually and (2) consider the combination of classifiers in one integrated classifier. The algorithms used for the system are MLP, SVM, deep learning and Naïve Bayes classifier. The experiments performed using Rapid Miner on the clinical database of ectopic pregnancies were collected from “Virgen de la Arrixaca” hospital in Spain. The reported results show that the SVM improves accuracy for both a single classifier and the three-stage classifiers. Moreover, SVM and MLP both performed better in terms of sensitivity, accuracy, and specificity. The system helps doctors to take their decision for initial treatment on time.

In low resource areas, facilities for sonographers and ultrasound machines are not easily accessible. To address the issue, the ML approach [69] was proposed for predicting the fetal weight at fluctuating gestational age. Ensemble models of ML including Random Forest, Extreme Gradient (XG) Boost, and Light GBM were used, as such models create multiple models to enhance accuracy of the approach. First, the parameters of the genetic algorithms were initialized; then, the optimization parameters were selected. Experiments were performed to predict the fetal weight, and the reported results show that the multiple model approach gives better results as compared to an individual algorithm in terms of accuracy.

Around the world, the effects of infertility are common in one out of seven couples. In vitro fertilization (IVF) is the best suggested treatment for such couples. The main concern of such patients is the successful results of IVF treatment, which mainly depends on the factors and number of attributes. For this purpose, automated classification with ML, integrating the hill climbing feature selection algorithm, is proposed [70]. To assess the prediction ability of IVF, 25 attributes and well-known ML algorithms including MLP, SVM, C4.5, CART and RF were used. Experiments were performed using MATLAB and the results show that the proposed approach gives better prediction accuracy than the existing techniques such as neural network-based image analysis method, and total recognition by adaptive classification experiments (TRACE). Prediction accuracy was measured in terms of well-known evaluation measures including F-measure, area under the curve (AUC) and accuracy.

Braxton Hicks, which is normally known as false labor or contractions during pregnancy, has been experienced by mostly women. Most women failed to identify such pains. For pain track analysis, a prediction model was proposed [71]. First, SVM is used by the model for sentiment analysis. Later, images are taken as input, and their facial expressions are extracted using facial recognition algorithm. To differentiate the pains such as false labor or true labor, SVM is used. The experiments were performed using MATLAB, and the results show that the model performs better as compared to existing ones in terms of accuracy.

### 4.3. Sensors Used in Infant Healthcare Systems

For infant health during pregnancy, sensors have created many possible ways to tackle complications such as infant motion, diabetes status of the infant, and heart rate (HR) of the infant during or after birth. These wireless and wearable sensing technologies provide continuous health monitoring of infants [72] and can update parents and doctors. Thus, by exploiting sensing technologies via different machines, android mobile applications, and instruments, all such issues including movements and respiratory motions of an infant can be controlled [22,73,74,75]. In short, it can be concluded that the inclusion of sensors in the medical field has helped many of professionals and doctors to have a balanced monitoring of the infant’s health status.

Monitoring the physiological health of infants is crucial. In the past, traditional methods were used where various equipment pieces were attached to the infant’s body, which caused discomfort to the subjects. In order to overcome this issue, a sensor-based bed was designed that monitors the physiological health of the infant [76]. The proposed technology exploits Ballisto cardiographs (BCGs), which provide heart mechanical activity and efficiently record physiological measurements. The proposed bed includes heart rate (HR) and breathing rate (BR) using load-cell sensors signaling. To validate the proposed bed, 13 experiments were carried out in which four infants participated. As a reference signal, a commercial device was used to measure electrocardiogram signals and breathing signs simultaneously. Then, the results of the load-cell sensors and reference signal were compared by using the proposed algorithm. The comparison verified that the proposed technology of BCG performed acceptably well for both HR and BR with an average error rate of 2.55% and 2.66%, respectively. Although promising results were recorded from the experiment, it has limitations such as minimum number of measurements and insufficient recording time. Future works can be conducted on these two limitations for better respiratory distress solutions for infants using the proposed method.

Fever, sneeze, and cough are some of the few common infant diseases. Among these three, fever is the one that fights against the infections that attack infants. In some cases, this fever can be dangerous for an infant’s health. Sometimes, infants have a severe increase in their body temperature for a short interval of time, which remains unnoticed for either their parents or caretaker. This sudden increase in body temperature may lead to severe injuries such as epilepsy. In this regard, a lightweight device was proposed [77] and developed in Malaysia to alert the infant’s parents about their abnormal body temperature. The device is small in size and comfortable to wear for an infant, and it continuously monitors body temperature. The wearable device has LM35 sensors in it, which gather the infant’s body temperature and send the information to parents via a wireless network. A micro-controller named Arduino ESPresso Lite 2.0 controls the LM35 sensor to measure temperature. After collecting the temperature data, the micro-controller ESPresso transfers and stores these data to the cloud server. Later on, the data are processed and sent via Wi-Fi interface to the mother’s or caretaker’*s* mobile phone. To establish a wireless connection for the Wi-Fi interface, an ESP8266 is used. The proposed device is easy to implement and is less expensive. This device can provide better communication between a mother and infant. For future work, more sensors can be exploited to the proposed device to give information about heart rate, oxygen saturation, etc.

In another study, vision sensors were used to examine babies’ behavior while employing the intelligent multimodal system [78]. The proposed method is different from traditional wearable devices that make babies uncomfortable when attached to their bodies. This vision sensor-based monitoring system uses control chart techniques. This control chart technique provides baby behavior in an analyzed manner. The proposed method’s control chart is constructed using Raspberry Pi (RPi)-attached vision sensors, which provide real-time frames. On the control chart, there are two points: (1) upper control limit (UCL) and (2) lower control limit (LCL). If the baby’s motion drops below LCL or the baby’s movement surpasses UCL, an alert is transferred via interconnected IoT devices to baby caretakers. Performance evaluation measures on collected data are entirely accurate and efficient, which show the proposed system’s success. The proposed system can be developed for home use with only one RPi, and a network can be made by using multiple RPi for a healthcare center or nursery.

Presently, many mothers are giving premature birth, due to which the infant faces many health issues. Health issues force the infant to stay in an incubator system for his survival such that that hospital management can monitor the infant manually all the time. The monitoring of healthcare is well improved through cloud computing, IoT and wireless medical sensors that result in accurate and continuous operations. To ensure that the doctors and incubator monitoring system’s accuracy and authenticity avoid impersonation and replay attacks, another system was proposed [79]. The objective of the system is to give a new encryption scheme for constructing efficient authentication protocol, and its implementation leads to continuous physical monitoring, resulting in a lessened need for nurses and doctors. The proposed protocol consists of various sensors to measure the infant’s essential parameter values that reduce the system’s cost and improve performance with all security measures. As mutual authentication is necessary between infant incubator monitoring systems for wireless data transmission, a new encryption scheme enhances computational efficiency through limited resources with balanced security information in the system.

The heart rate (HR) of newborn matters a lot for monitoring future health. If a newborn baby’s breath does not start, it may cause a reduction in heart rate. In addition, it badly affects the circulation of blood to the baby’s organ. At the time of birth, the neonatal staff manually records the baby’s HR by listening to the baby’s heart. This technique is neither efficient nor accurate. To overcome this issue, a novel device is proposed [80]. The proposed device uses a smart mattress with electrometer-based amplifier sensors and the screen-printing technique. The device records and monitors the breathing and electrocardiogram of a baby. To illustrate the suitability of an electrocardiogram (ECG) monitoring-based smart-mattress, many concept tests were performed and proven. According to the reported results, the device is accurate and quickly gives ECG readings of a young infant in less than 30 s. This novel development has potential to help neonatal staff in the resuscitation procedure for newborn babies and the delivery room for the newborn baby’s stabilization. The proposed device reduces the times for the assessment of resuscitation process success. The device performance is four times quicker than pulse oximetry.

Epilepsy is becoming a common disorder currently, causing sudden seizures or sensations, awareness loss, and unusual behavior. This neurological disorder affects any human being of any age. There has been a dire need for seizures’ early detection, as it may improve treatments and give timely warning to the patient. In this regard, a system is proposed to assess the utility of ANS metrics for the identification of early seizures to delineate the period of preictal in terms of specificity and sensitivity [81]. This system uses wearable devices such as an automatic nervous system (ANS), which offers promising results. It is easy to use and is cost-effective as compared to impractical EEG. The investigation included 66 people with epilepsy who have a continuous recorded unique dataset of multi-day wristband data and statistical testing with seizure surrogate data, including temperature, heart rate, and electrodermal activity that did not exhibit consistent trends. These investigations result in differences between the preictal and inter periods regarding these signals’ entropy, variance, and mean, and they potentially afford to search for more personalized seizure makers. Electronic design automation (EDA) signal entropy was used to increase seizures in a small subset of patients, whose findings may give deep insight into epileptic seizure pathophysiology with respect to ANS function. Through these wearable devices, the detection of changes becomes easy and more feasible. Therefore, it provides the latest opportunities in seizure risk evaluation and forecasting based on easy-to-use and non-invasive devices.

Human detection regarding human health and development through high resolution and sensitivity image sensors is common in the present era. However, these biological and environmental sensors are expensive and demand powerful processing capability. Thus, it is undoubtedly challenging to analyze humans during their regular daily life routine at home. Coping with these challenges, a detection system is proposed that uses low-cost infrared (IR) technology-based location, thermal environment, motion, and temperature sensors [82]. It is beneficial for long-term evaluation in the home environment. This latest technology is tested to visualize the thermal environment and parental care effects on the common marmoset known as circadian rhythms. First, a comparison of this system is made with a manual analysis technique for validating the design. Afterward, circadian rhythms are assessed from the postnatal day in the standard four marmosets. In the circadian phase, patterns of age-dependent shifts are shown through the biological indices of body surface temperature and locomotion velocity. The development of these circadian rhythms and principal analysis of components was affected by environmental variables. A novel basic pattern of bipolar disorder—blood treatment—blood test (BD-BT) correlation is revealed by signals superimposing imaging methods in correlation with day/night animal switching that is older than a postnatal day, which is also one of the limitations of this study. The switch origin is associated with BT and BD rhythms around earlier times of feeding. In the future, this technique has value for understanding care conditions in which non-invasive home monitoring is beneficial and useful, which also further suggests the potential in adapting this technique that well facilitates home AI program implementation and development for healthy development support.

Wearable sensors play an important role in the field of medicine because these sensors can conveniently give important physical information in real time without bulky instruments. However, these sensors have many drawbacks such as frequent battery charging cycles and user inconvenience. These drawbacks can be removed after reducing power consumption, which arises due to the battery’s capacity or size. To overcome the drawbacks, an ultra-low power sensor was proposed [83], in which a signal repeater and a particular switch were introduced that significantly reduce the use of power, as both provide power when there is a dire need. Afterward, the characteristics of antenna radiation were observed, which are an essential factor in the wearable sensors. The soft encapsulation method improved the antenna radiation that maintains good wear ability in daily life, which can be degraded by the polymeric encapsulation layer. The human body’*s* safety is also verified in this proposal through radio frequency (RF) and absorption rate simulations. Furthermore, the infant’s sleep position was monitored in the wearable sensor part by an accelerometer sensor. As infants cannot communicate well and cannot talk about their harmful situations, it is also challenging for caregivers to keep an eye on them; thus, this infant sleep monitoring system helps the caregivers be alert in the infant’s unsafe situations.

The framework of locomotion is displayed by the infant’s earliest motor skills that determine developmental progression. Conversely, motor dysfunction negatively impacts physical insight, spatial, balance, and cognizance, and it is considered significant in infants’ autism spectrum disorder (ASD). ASD refers to an infant’s sitting and standing delay and head lag. A system is proposed in [84] that uses a wireless device known as opal sensors, comprising a 3D magnetometer, 3D gyroscope, and 3D accelerometer to study the full-day HR of the infant’s movement. Motion complexity in an infant is measured through these sensors, which is important for normal motor development, as less body movement reflects ASD symptoms. These lightweight sensors report only 14 bit resolution and have a range of 6 g that results in recording at 20 Hz. The data are synchronized from both the left and right sensors and are stored in the individual sensor’s internal memory. At every visit, the data can be downloaded. The data records include the infant’s sleep and wake status. ASD outcome and motion complexity have the most vital relationship with each other over adaptive skills and cognitive ability results. Primary motor development measuring objectives are required to identify a typical infant performance of motor that can show later ASD risks. Early infant’*s* motor development can easily be tracked by motion complexity and removes the risk of ASD in HR infants. Table 4 provides an overview of the latest models proposed in recent years as discussed above.

### 4.4. AI/ML Techniques Used in Infant Healthcare Systems

In recent years, ML and deep learning has become an emerging technology for the various health monitoring systems to monitor heart rate, glucose level, blood pressure and growth of infants. Algorithms such as: ANN, SVM, CNN, Random Forest and others are widely applied for the timely diagnosis of the disease [48]. Table 5 gives an overview of the latest models.

To enhance the health and behavioral monitoring system, a new method was proposed in [85]. Gaming data and body parameters were used to analyze the health of child. For large scale data management and comparison, Hadoop is used, whereas C4.5 and ID3 algorithms are used for classification. A class label represents different types of disorders such as disruptive behavior disorders and attention-deficit/hyperactivity disorders. The reported results show that the predictions made using C4.5 achieve better results than the ID3 algorithm in terms of accuracy and execution time. In the future, more mind games can be added, and using more body sensors may help to monitor the physical and psychological state of the child at an early time.

The timely diagnosis of pneumonia is rare in developing regions. Diagnosis through the ultrasound of lungs is useful but has some limitations, as it requires a trained person. To overcome the limitations, an automatic classification approach for pneumonia was presented in [86]. First, the skin and tissues were eliminated from the ultrasound frames of lungs. Then, the frames were analyzed using ANN. To differentiate the healthy lungs from the infected ones, 60 frames of ultrasound were used. ANN is trained to correctly identify the infiltrates of pneumonia. The results show that the proposed approach achieves 90% sensitivity and may be used to develop the operator independent system for timely diagnosis.

In [87], to predict the pediatric mortality in ICUs, an ML-based model was developed. The models named as Pediatric Risk of Mortality Prediction Tool (PROMPT) and Convolutional Neural Network (CNN) was used to develop the model. It consists of two layers: the first layer has one-dimensional convolutional operations, and the second layer has max pooling. Performance of PROMPT was compared with other ML algorithms such as Long Short-Term Memory (LSTM), and Gradient Boosting Decision Trees (GBDT). The results showed that the developed tool achieves high sensitivity and that the tool has a better ability to predict patients at a high risk of mortality.

For physicians to monitor the patients’ health at a distance and take necessary actions on time in the case of an emergency, a smart health monitoring system was developed [88]. First, health parameters such as ECG, heart rate, blood pressure, fall detection and temperature were sensed using body sensors. The SVM is used as a classifier to classify the sensed data and then generates an alert if an emergency is occurring. It also informs the ambulance, if required. The developed system has been implemented in rural areas to connect patients with specialist doctors in big cities and hospitals for timely treatment.

## 5. Datasets

This section represents the real-world datasets used for the experimentation purpose in maternal and infant healthcare systems. The primary intent to review these datasets is to discuss the nature and attributes considered in healthcare systems for prediction, since both nature and the attributes of the dataset have profound impact on the selection of ML algorithms and prediction results. The datasets include infant images to monitor baby position [89], MRI database [90,91,92,93], infant mortality, etc. The nature of datasets covers images, videos, and text-based data [92]. Table 6 presents the summarized view of publicly available datasets.

The Pregnancy Risk Assessment Monitoring System (PRAMS) is designed to monitor and identify the risk factors that occur before, during and after pregnancy. Participants were selected randomly based on birth certificates from New York City. The dataset size is 1.11 KB and is freely available for research purposes. The infant mortality dataset contains the counts of death of infants, those which are less than 1 year old based on the death certificates in New York City. The rate was calculated by dividing the number of infant deaths by the counts of live births. The dataset size is 2.44 KB and is freely available for research utilization.

UNICEF together with their key partners are analyzing the health status for mother and child. They have different collections of data such as antenatal care, newborn care, delivery care and maternal mortality [93]. UNICEF also contains the child mortality data of different stages such as still birth, neonatal mortality, and under-five mortality [23,94]. Data were collected from different countries. Maternal and child health data and statistics [95] are contained in a digital library which provides different datasets about births, infant deaths, child care, and indicators of child including stages of maternal and child. The website has a variety of toolkits to analyze the data and statistics online as well as provides statistics about adolescents, pregnant women, infants, and their families.

## 6. Research Opportunities and Challenges

In the relevant literature, on sensing-based healthcare systems in general and maternal and infant healthcare systems specifically, the research studies report different research challenges that open new research opportunities for researchers in this domain. This section presents a discussion on these challenges and research opportunities and describes them in four sections: (a) application of state-of-the-art techniques related to ML and AI in maternal and infant healthcare systems; (b) issues related to availability of maternal and infant real-world datasets; (c) lack of advance IoT-based monitoring systems; (d) research issues associated with IoT-based healthcare systems.

### 6.1. Application of State-of-the-Art Techniques ML/AI in Maternal and Infant Healthcare Systems

To accurately predict the diseases of mothers and infants, different health complications need to be considered in association. For instance, predicting the congenital anomalies in the fetus can be optimized if risk factors such as pre-eclampsia can be associated with other pregnancy complications [67]. There is a need to study different diseases of mothers and infants in association such that better prediction models can be designed. In addition, diseases related to psychological disorders in pregnancy using AI techniques should be investigated [96]. Similarly, to monitor the behavioral changes in children, more mind-related games can be considered [85]. The literature highlights psychological and behavioral issues to be considered along with medical diseases. In this regard, mobile apps for existing developed systems can be created that are compatible with different operating systems such as MacOS, to facilitate more people [66], and AI-based techniques may be utilized to develop an independent operating system for both infants and pregnant women [86]. Since prediction accuracy of a model is desirable, time series of historical data of patients for physical examination should be exploited [69]. In addition, for enhancing the accuracy of complex indicators, other ensemble classifiers need to be studied specifically related to urgency, such as the admission of pregnant women and infants in intensive care units. In addition, for maternal and infant healthcare systems, deep learning algorithms such as Convolutions Neural Network, Recurrent Neural Networks and Long Short Term Memory are recommended to be used in future research studies, as these techniques are widely used in other areas, and their performance in terms of accuracy and other standard research evaluation measures for classification have proven to be effective. Lastly, the labeling of data for deep learning and ML is time-consuming and expensive. The supervised learning-based algorithms need proper refinement and improvement to enhance pregnancy outcomes, and such techniques can be incorporated for real-world clinical care [31,87].

### 6.2. Lack of Availablity of Real-World Maternal and Infant Datasets

Similar to in other healthcare systems, the availability of real data of mothers and infants from health institutions is critical because patients do not allow to publicize their personal data and reports. It is sensitive in nature, and hackers can easily modify it, which leads to wrong treatment or diagnosis of diseases and may cause increases in mortality rate [47]. In addition, this situation urges new researchers to develop their own datasets for maternal and infant healthcare systems. In such a scenario, it is recommended to develop anonymous data from the original datasets such that they can be shared with researchers for future research studies. In addition, even if the data are not anonymized, the researcher should be trained to anonymize them before use and before making it public. In this way, safe and easy availability of data can be ensured. In addition to the text data, images such an ultrasound, MRI images and video datasets such as videos of activity of newly born babies are of equal significance in maternal and infant healthcare systems. However, in the case of video datasets, it is observed that the duration for video recording to monitor the variety of movements during sleeping or waking is short in length. Consequently, the duration of the video should be increased, especially to monitor the health conditions of the infants [76]. Moreover, accurate labeling of data in sensitive maternal and infant healthcare systems allows medical experts to be engaged for labeling of class attributes because it ultimately influences the accuracy of prediction.

### 6.3. Need of Advanced Sensory and Monitory Systems Using IoT

Numerous applications of sensors such as BP sensing, heart beat monitoring, and glucose level sensing with computing technologies bring great improvements in healthcare systems. It is also recommended to use sensory and monitoring systems for maternal and infant healthcare systems. For instance, evaluation of perinatal outcomes in pregnant women may be a future challenge that can reveal further confusion in such conditions such as normoglycemia [19], and sensory technology can a play crucial role in this regard and may help to reduce mortality rate. Similarly, sensor-based beds can be used to solve the respiratory distress problem in infants [76]. In addition, to monitor fetal movement, a wearable belt needs to be designed and developed with optical fiber sensors that can reduce motion artifacts and classify movement type [60]. The relevant literature recommends exploring more sensors that may be used to monitor heart rate and oxygen saturation along with temperature [77]. Such issues can be potential research work in the future. In addition, a comparison between the existing systems and the IoT-based smart system is crucial to monitor the reliability issue, area of application, implementation techniques and technology [46].

### 6.4. Research Issues Raised Due to IoT-Based Systems

Challenges including data management, security, and energy management, etc., need to be sorted out in IoT-based healthcare systems [49] in general and maternal and infant healthcare systems specifically. Managed data need to be stored and processed properly, as large amounts of data are generated and processed. It could be addressed if any company gives enough storage to manage all data properly. As security protocols are a challenging task, devices are linked to the internet in different environments, which cause issues for the compatibility of network security. Security challenges may be solved by motivating the users to practice built-in features for devices. In addition, the encryption of data at the source device may address these issues in maternal and infant healthcare systems. As far as energy optimization is concerned, medical sensing devices need a continuous supply of energy. Since 24/7 monitoring without unwanted disruption is desirable, energy optimization is an important concern in sensing devices. In the maternal and infant healthcare systems, where continuous monitoring of mother and infant is desirable, state-of-the-art techniques and approaches need to be introduced for data management and security and energy optimization.

## 7. Conclusions

With the rapid advancement in technology and sensor-based architectures during the last decade, health care systems have been developed to meet the needs of patients from remote and rural areas that are deprived of timely medical assistance. Wireless technology has made it possible to remotely monitor mother and infant health conditions using sensors to detect any possible problem in time. Significant developments in recent years in AI has enabled such health care systems to predict and assist medical practitioners in diagnostics. IoT-based applications provide fast and accurate diagnosis of diseases to overcome challenges such as storage, cost, latency and many more.

In this study, maternal and infant-based health care systems are reviewed and classified based on sensors and ML techniques that are presented in the latest literature. Sensors that are used as wearables and IoT devices for health monitoring are explored and summarized by providing salient features and specific diseases of mother and infants for which they are designed. Similarly, intelligent and predictive algorithms are presented for a thorough understanding of state-of-the-art research work in this area. Surveys provide an overview of smart health care services, alert generation using computing platforms that include cloud, fog computing, and edge computing. A variety of real-world datasets and their details in terms of attributes, size, format, etc., are presented, which help the researchers for choosing a dataset for their research study. It also explores the fact that we lack availability of real-world datasets in this area. In further extensions, we aim to include classifications in terms of state-of-the-art computing platforms. Potential future work directions for researchers in the relevant domains are also highlighted. In addition, the research challenges and opportunities that researchers face are identified. Such research challenges are potential research topics for future researchers in the field of healthcare systems in general and for healthcare systems targeting monitoring of mothers and their infants.

## Figures and Tables

**Figure 1 sensors-22-04362-f001:**
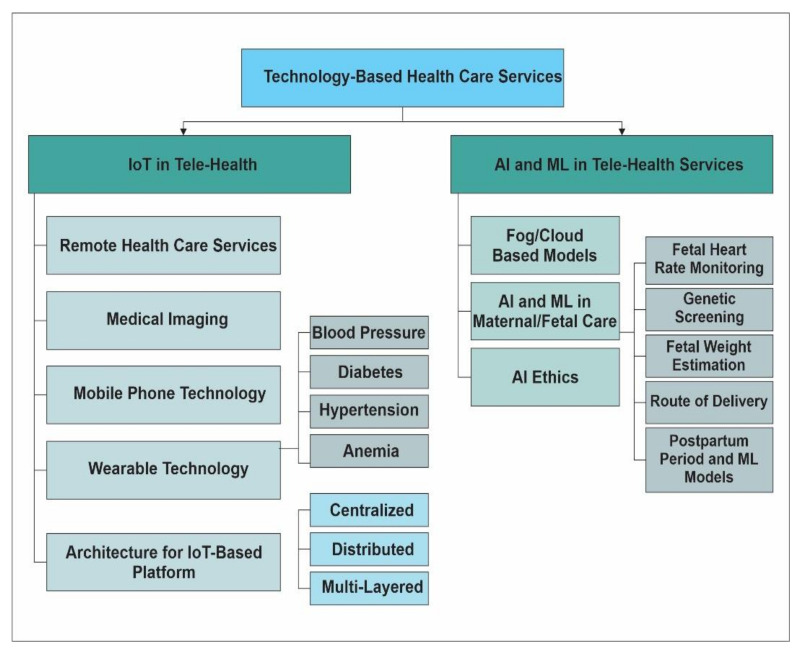
General classification of healthcare services.

**Figure 2 sensors-22-04362-f002:**
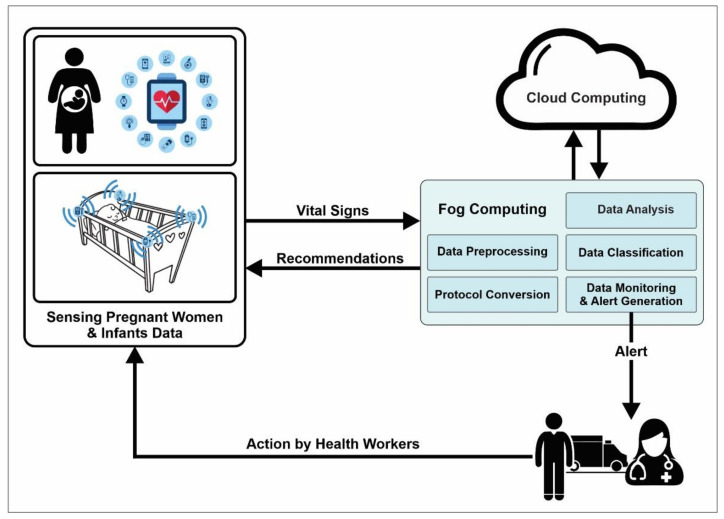
Architecture of healthcare systems.

**Figure 3 sensors-22-04362-f003:**
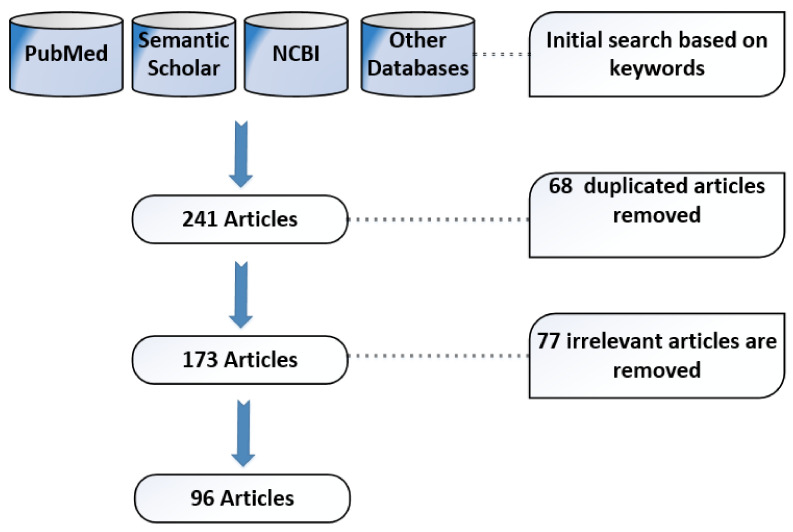
Resource databases and article screening.

**Figure 4 sensors-22-04362-f004:**
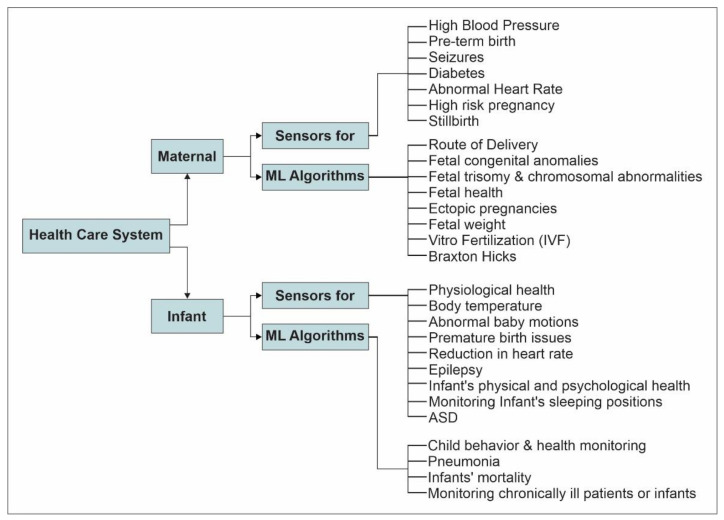
The structure of review.

**Figure 5 sensors-22-04362-f005:**
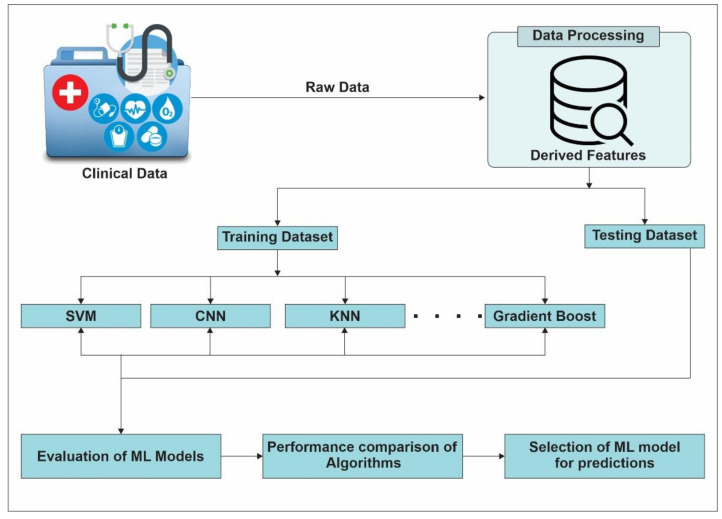
General framework to predict patient’s health status using ML.

**Table 1 sensors-22-04362-t001:** A comparison of this review with existing review papers.

Sr. No.	Ref. (Year)	Review Methodology	Contribution	Major Difference from This Review
1	[7] (2020)	Systematic review and meta-Analysis	A study on the effectiveness and safety of pharmacologic interventions for the treatment of retained placenta.	No AI methods and sensing devices are reviewed. Only one maternal problem (retained placenta) is considered.
2.	[30] (2019)	Seminal review	Identifies key characteristics and drivers for market uptake of ANN for healthcare organizational decision making to guide further adoption.	No focus on maternal and infant issues. Sensing devices are also not part of this study.
3	[31] (2021)	Systematic review	A systematic review on the ways that AI and ML including deep learning methodologies can inform patient care during pregnancy and improve outcomes.	Sensing-based healthcare and infant health issues are not considered in this review.
4	[32] (2020)	Systematic review	A study to provide an evidence map of the current available evidence on ML in pediatrics and adolescent medicine.	Maternal issues are not a focus. Furthermore, a sensing-based remote patient monitor is not included in this review
5	[48] (2020)	Systematic review	The main characteristics and outcomes of studies using Computerized Decision Support (CDS) and ML are demonstrated, to advance our understanding toward the development of smart and effective interventions for childhood obesity care.	Maternal issues are not a focus, and only child obesity is the scope of review.
6	[49] (2019)	Systematic review	A systematic literature review protocol to study how mobile computing assists IoT applications in healthcare is presented.	Maternal and infant healthcare solutions are not studied
7	[50] (2021)	Nonsystematic comprehensive survey	Review covers AI-based algorithms for novel prediction models, better diagnosis, early identification, and monitoring of women during pregnancy, labor, and postpartum to advance research, clinical practice, and policies, and ensure optimal perinatal health.	No focus on infant issues, and sensor networks are not considered for the monitoring of maternal and infant healthcare
8	[51] (2018)	Comprehensive review	Authors outlined the medical applications, ethical and international standardization challenges about the 5G e-health systems.	AI is not the primary concern in this review. Moreover, it does not specifically target pregnancy and infant-related problems

**Table 2 sensors-22-04362-t002:** Sensors used in maternal healthcare systems.

Sensor Based System	Features	Year	Disease	Dataset Acquisition
Wearable Technology, a solution to hypertension during pregnancy [55]	VO7 wearable model	2018	High Blood Pressure, Pre-term birth	From mobile application
Eclamptic Seizures monitoring by wireless sensors network [56]	5G wireless sensing system	2019	Seizures	Not available
Nanocube-based flexible sensors for detection of hemoglobin and glycated hemoglobin [57]	Electrochemical sensors comprising double imprinted nanocubes	2019	Diabetes	Blood samples of healthy and diabetic pregnant women
Invu System: Home fetal and maternal heart rate monitoring [58]	Wireless electrical and acoustic sensors	2020	Abnormal heart rate of mother and child	From 147 participant women
Normoglycemia and GDM in early pregnancy through a continuous glucose monitoring system [19]	Continuous glucose monitoring system	2020	Diabetes	From 96 participant women
Measurement of fetal hemodynamics and evaluation of health factors through intelligent ultrasound sensors [59]	Intelligent ultrasound sensors	2020	Diabetes	From questionnaire
IoT platform for smart maternal healthcare using wearable devices and cloud computing [46]	IoT-based platform with wearable devices and cloud computing	2021	High risk pregnancy	From questionnaire
Use of optical fiber sensors for fetal movement counting [60]	Optical Fiber Sensors	2021	Stillbirth	From 3 volunteers

**Table 3 sensors-22-04362-t003:** Summary of AI/ML techniques used in maternal healthcare systems.

AI/ML Based Systems	Year	ML Algorithms Used	Disease	Dataset/Availability
Computerized Prediction System [65]	2018	ANN	Route of delivery	Data consist of 2127, 3548 and 1723 deliveries for the years 1976, 1986 and 1996/No
Fetal Health status Prediction using ML [66]	2018	Logistic Regression, Locally Deep SVM, Neural Networks, SVM, Averaged Perception, Decision Jungle, Decision Forest, Bayes Point Machine, Boosted Decision Trees	Fetal congenital anomalies	Clinical database of 96 pregnant women/No
Two-stage approach using computational Intelligence System [67]	2018	ANN	Fetal trisomy and other chromosomal abnormalities	Dataset comprises 72,054 euploid pregnancies/No
Deep CNN Regression Model for 3D Pose Estimation [68]	2019	CNN	Fetal health	MRI scans of 40 newborns and 93 reconstructed MRI scan of fetus/NO
Decision Support System [1]	2019	Multi-Layer Perception (MLP), Deep learning, SVM, Naïve Bayes classifier	Ectopic pregnancies	406 cases of ectopic pregnancies collected from “Virgen de la Arrixaca” hospital in Spain/No
Prediction of Fetal Weight using Ensemble Learning [69]	2020	Random Forest, XG Boost, Light GBM algorithm Genetic algorithm	Fetal weight	Dataset comprises 4212 intrapartum recordings/No
Machine learning approach for IVF treatment [70]	2020	Multi-Layer Perception (MLP), SVM, C4.5, CART, RF	In vitro fertilization (IVF)	Data from infertility clinic in Istanbul/No
Pain Track Analysis [71]	2021	Facial recognition algorithm accompanied by SVM, Decision tree	Braxton Hicks	Database of images/No

**Table 4 sensors-22-04362-t004:** Sensors used in infant healthcare systems.

Sensors Based Systems	Features	Year	Disease	Data Acquisition
Load-cell sensors based physiological signal monitoring bed for infants [76]	Load-cell signal sensors	2016	Physiological health	4 infant patients
Body temperature monitoring of infant using IoTs [77]	LM35 sensor	2018	Body temperature	Not available
Use of vision sensors in IoT for intelligent baby behavior monitoring [78]	Vision sensors	2019	Abnormal baby motions	Baby’s video
Computationally efficient mutual authentication protocol for remote infant incubator monitoring system [79]	Wireless medical sensors	2019	Premature birth issues	Not available
Cardiac monitoring of babies through non-invasive smart sensing mattress [80]	Electrometer-based amplifier sensors	2019	Reduction in HR	Concept tests
Autonomic nervous system changes detected with peripheral sensors in the setting of epileptic seizures [81]	Peripheral sensors	2020	Epilepsy	66 patients participated for the study
Inexpensive Home Infrared Living/Environment Sensor with Regional Thermal Information [82]	Infrared sensors	2020	Infant’s physical and psychological health	Not Available
Ultra-Low Power Wearable Infant Sleep Position Sensor [83]	Switch sensors	2020	To monitor infant’s sleeping positions	24 infants
Measurement of infant complex motions using wearable sensor technology [84]	Wearable opal sensor technology	2021	ASD	5 infants

**Table 5 sensors-22-04362-t005:** AI/ML techniques used in infant healthcare systems.

AI/ML Based Systems	Year	ML Algorithms Used	Disease	Dataset/Availability
IoT based child behavior and health monitoring system [85]	2017	C4.5, ID3 algorithm Decision Tree	To monitor child behavior and health	Data are self generated/No
Automatic Classification of Pneumonia using ANN [86]	2018	Artificial Neural Network (ANN)	Pneumonia	60 ultrasound images/Yes https://osf.io/hmr3w/ Access Date: 11 May 2022
PROMPT [87]	2019	CNN	Infants’ mortality	1977 patients/No
ML based Health monitoring system [88]	2020	SVM	To monitor chronically ill patients or infants	Data collected through body sensors/No

**Table 6 sensors-22-04362-t006:** Summary of publicly available datasets.

Datasets	Source	Attributes	Format	Language
Pregnancy Risk Assessment Monitoring System (PRAMS)	https://data.cityofnewyork.us/d/rqgf-94xs Access Date: 6 February 2022	Year, source, question, prevalence %, lower 95% confidence interval, upper 95% confidence interval	csv	English
Infant Mortality	https://data.cityofnewyork.us/d/fcau-jc6k Access Date: 9 February 2022	Year, maternal race, infant’s mortality rate, neonatal mortality rate, post neonatal mortality rate, infant death, neonatal infant death, post neonatal mortality rate, No. of live birth	csv	English
Baby Monitor Forecast	https://www.kaggle.com/c/fiap-fsbds-baby-monitorforecast/data?select=test.csv Access Date: 11 February 2022	Id, date, mes, weekday, mergem, Venda, desconto, outdesc, outmg	csv	English
Maternal and Child Health Data of UNICEF	https://data.unicef.org/resources/dataset/maternal-health-data/https://data.unicef.org/topic/child-survival/ Access Date: 23 February 2022	Country, year, mothers’ age, source	csv	English
Neuro Developmental MRI Database	https://jerlab.sc.edu/projects/neurodevelopmental-mri-database/ Access Date: 23 February 2022	Age, 1.5 T, 3.0 T, total, notes	Tar.gz	English
Infant Death Dataset	https://www.cdc.gov/nchs/fastats/birth-defects.htm Access Date: 23 February 2022	No. of infant deaths, infant deaths per 100,000 live births, cause of infant death	pdf	English

## Data Availability

The datasets used during the current study are available on reasonable request.

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
