# Peer review of "Sensing and Artificial Intelligent Maternal-Infant Health Care Systems: A Review"

_sensors, 2022, doi:10.3390/s22124362_

Round 1

Reviewer 1 Report

This article is a comprehensive literature review of remote sensing and cloud processing incorporating  ML/AI technologies to improve maternal/fetal health with monitoring and clinical decision making . It is an ambitious goal but I comment the authors to provide a good overview of what has been accomplished. For ML I agree that most clinically relevant decisions particularly with multi set data SVM has always shown superior delineation. I think incorporating all these technologies into reality would require infrastructures that would allow efficient wireless networks at the level that can support sophisticated cloud technologies that  can process and encrypt large data computing. Also privacy and legal ramifications are region dependent as well. Nevertheless I enjoy reading this article and would recommend publicly as is.

Author Response

We are thankful for your valuable comments for the improvement of the article. We agree with your point of view. We have improved the article for a detailed and good overview of the content is presented.

Reviewer 2 Report

  1. High-definitional figures are required in scientific writing.
  2. Some acronyms are used without related full name definitions at their first appearance.
  3. The motivation should be elaborated in the introduction section. Currently the reviewer cannot pick out this since this study is a combination of existing studies, which means, there is seldom original idea from the authors.
  4. There is no correspondence author in this submission.
  5. The abstract section is too long. We generally only need about 150 to 200 words here.
  6. The contributions should be There are some state-of-the-art studies about the ICT-assisted health and diagnosis. The authors should include the latest studies and better to compare this work and the existing ones to clarify the novelty of this study, for instance, “Perinatal health predictors using artificial intelligence: A review,” in Womens Health, “Design and Implementation of 5G e-Health Systems: Technologies, Use Cases and Future Challenges,” in IEEE Communications Magazine, “ML-Net: Multi-Channel Lightweight Network for Detecting Myocardial Infarction,” in IEEE Journal of Biomedical and Health Informatics, etc.
  7. The logical structure is chaos. The authors may want to polish the presentation and use some summary/remark to highlight the main points from each section that the readers can take away.
  8. What is the relationship between AI-based maternal healthcare system and AI-based Infant healthcare system? Additionally, what they can contribute to the AI maternal-infant healthcare system as the title claimed? Currently these two parts are separately discussed.
  9. Are the datasets obtained from the authors’ studies or not? If not, what aspect can they be used to contribute to this article?

Author Response

Comment 1:

High-definitional figures are required in scientific writing

Response:

As per your comment the figure with low resolution are improved and incorporated in the updated version of the paper.

Please see. Figure 2, 4, 5

Comment 2:

Some acronyms are used without related full name definitions at their first appearance

Response:

Thanks for highlighting the undefined acronyms. The manuscript is proofread to identify the undefined acronyms. Following is the list of such acronyms; these are defined in the manuscript. Similar errors are also removed.

Line #

Acronym

Full form

40

C-section 

Cesarean Section

59

IoTs

Internet of Things

76

NG tubes

Nasogastric tube

134

AI/ML

Artificial Intelligence/Machine Learning

178

SAP HANA database

Systems, Applications and Products High-Performance Analytic Appliance database

185

GUI

Graphical User Interphase

390

XG Boost

Extreme Gradient Boosting

495

ECG

Electrocardiogram

515

EDA

Electronic Design Automation

526

IR technology

Infrared

535

BD-BT correlation

 Bipolar Disorder- Bleeding Time correlation

Comment 3:

The motivation should be elaborated in the introduction section. Currently the reviewer cannot pick out this since this study is a combination of existing studies, which means, there is seldom original idea from the authors.

Response:

We are thankful to point out this deficiency. We have rewritten the last paragraph of introduction section to highlight the motivation behind this review.

Please, view Section1 Introduction, page 4.

In addition a separate Section 2 Rationale is included to clarify the motivation and rationale for this review paper. This section also includes Table 1 that summarizes the existing relevant review papers and major differences with this review, which clears the novelty of the presented review.  

Please view Section 2 Rationale, page 5.

Comment 4:

There is no correspondence author in this submission.

Response:

Dr. Ehsan Ullah Munir is the corresponding author that can be viewed in the manuscript as well.

Please, view page 1

Comment 5:

The abstract section is too long. We generally only need about 150 to 200 words here.

Response:

The abstract is revised, and the length is reduced within the required word count. 

Please, see page 1, Section Abstract.

Comment 6:

The contributions should be There are some state-of-the-art studies about the ICT-assisted health and diagnosis. The authors should include the latest studies and better to compare this work and the existing ones to clarify the novelty of this study, for instance, “Perinatal health predictors using artificial intelligence: A review,” in Womens Health, “Design and Implementation of 5G e-Health Systems: Technologies, Use Cases and Future Challenges,” in IEEE Communications Magazine, “ML-Net: Multi-Channel Lightweight Network for Detecting Myocardial Infarction,” in IEEE Journal of Biomedical and Health Informatics, etc.

Response:

The papers are mostly selected from last 3 years to compile this review. However, we have reviewed the suggested articles by the respected reviewer and cited in the paper. A table is also included (Table 1) to compare this work and the existing ones to clarify the novelty of this study as discussed in the response of comment # 3.

Please view Section 2 Rationale, page 5.

Comment 7:

The logical structure is chaos. The authors may want to polish the presentation and use some summary/remark to highlight the main points from each section that the readers can take away.

Response:

Thank you for your valuable comments, for clarity and simplicity, advance organizer provided at the end of introduction section is improved; in addition, a preamble of the section 4 (it was section 2 in the previous version of the paper) is revised and improved.

Please see last paragraph of Section 1 Introduction, page 4 and first paragraph of Section 4, page7.

Comment 8:

What is the relationship between AI-based maternal healthcare system and AI-based Infant healthcare system? Additionally, what they can contribute to the AI maternal-infant healthcare system as the title claimed? Currently these two parts are separately discussed.

Response:

AI-based maternal healthcare system and AI-based Infant healthcare system are two separate systems; therefore, both are discussed separately in different sections. There is no relationship between both systems we have reviewed the related survey papers and to the best of our knowledge there is no review based on both systems together. The idea is to present AI-based systems of two different target patients in one review paper.  

Comment 9:

Are the datasets obtained from the authors’ studies or not? If not, what aspect can they be used to contribute to this article?

Response:

The datasets discussed in the article are used and reported in the cited articles. Since variation in selection of attributes may influence the prediction results, consequently this review article covers existing datasets and their attributes considered in different studies. This discussion will also help reader and researcher to know about the significance of different attributes in different medical healthcare systems. For further clarity, the authors have updated the last columns of Tables 2, 3, 4 & 5.

Please view Tables 2, 3, 4,& 5

Reviewer 3 Report

In general, the review methodology, the analysis structure, and the paper’s novelty are very unclear. The authors need to clearly address the review comments below before I reevaluate the whole manuscript.

[Comment 1] Novelty

The authors must compare their review with other existing review papers. Please state what is novel in this manuscript, in contrast to those other studies. I suggest the authors compare this manuscript with other existing review papers in a table for clarity.

[Comment 2] Review methodology

[Subcomment 2a] The research methodology and analysis procedure are unclear. At the end of Section 1, please clearly present how the authors review the studies, e.g., on what kind of flow the authors aim to observe them, etc.

[Subcomment 2b] The authors must also clearly state their review method, e.g., year span used for searching the studies, the publication source databases, the review steps, the number of studies obtained in each review step, etc. I suggest the authors use PRISMA to provide their statement (prisma-statement.org/PRISMAStatement/FlowDiagram). Such a clear review methodology is important to allow next researchers to reproduce the result and conduct further researches.

[Subcomment 2c] At the early part of Section 2, the authors must explain the differences between sensors-based and AI/ML-based systems. I could argue that sensors are used to collect the data, then the data could be processed using AI/ML methods. In other words, both classifications are related and might not be able to be separated from each other.

[Comment 3] Results

[Subcomment 3a] In all tables, I suggest the authors add information whether the datasets are provided or not, so other researchers can access the datasets they need (instead of only stating some of them in Table 4).

[Subcomment 3b] Section 4 is the most important part of the paper. The authors must focus on the findings for maternal infant cases, specifically, considering that there are already many studies on sensors and AI/ML for general healthcare. The following questions must be answered well: which findings are specifically new for the maternal infant cases? Sections 4.2 and 4.4 are still highly related to general healthcare and I believe could be found in many other general healthcare papers.

[Subcomment 3c] When stating the analysis points in Section 4, the authors must always relate to the findings they had in previous sections.

[Comment 4] Writing quality and clarity

[Subcomment 4a] The citation numbering must be written in an ascending order. In the manuscript, the authors jump from [16] to [21].

[Subcomment 4b] Instead of writing "..............." in the tables, please directly write its intended meaning, e.g., "not available", etc.

Author Response

Comment 1:

The authors must compare their review with other existing review papers. Please state what is novel in this manuscript, in contrast to those other studies. I suggest the authors compare this manuscript with other existing review papers in a table for clarity.

Response:

We are obliged for the valuable comment. A comparison of this review and existing relevant review papers is presented in the Table 1, which clearly depicts the major differences and novelty of this review.

Please view Section 2 Rationale, page 5.

Comment 2:

The research methodology and analysis procedure are unclear. At the end of Section 1, please clearly present how the authors review the studies, e.g., on what kind of flow the authors aim to observe them, etc.

Response:

The research methodology and analysis procedures are described in detail in newly added section 3.

Please view Section 3 Methods, page 6.

In addition, at the end of section 1, the advance organizer is revised and improved to present the flow.

Please view Section 1 Introduction, page 4.

Comment 3:

The authors must also clearly state their review method, e.g., year span used for searching the studies, the publication source databases, the review steps, the number of studies obtained in each review step, etc. I suggest the authors use PRISMA to provide their statement (prisma-statement.org/PRISMAStatement/FlowDiagram). Such a clear review methodology is important to allow next researchers to reproduce the result and conduct further research.

Response:

Thank you very much for the valuable comment, material and methods followed in this review are described in detail in the newly added section 3. 

Please view Section 3 Methods page 6.

Comment 4:

At the early part of Section 2, the authors must explain the differences between sensors-based and AI/ML-based systems. I could argue that sensors are used to collect the data, and then the data could be processed using AI/ML methods. In other words, both classifications are related and might not be able to be separated from each other.

Response:

We agree with the reviewer, it is so, due to the heading/content “Sensors based maternal system”, “AI/ML based maternal system”, “Sensors based infant system” and “AI/ML based infant system” that are misleading for the reader. Therefore, the phrases are replaced throughout the paper as under,

Sensors based maternal system   ……………… Sensors used in maternal system

AI/ML based maternal system ……………… AI/ML Techniques used in maternal system

Sensors based infant system…………………. Sensors used in infant system

AI/ML based infant system……………………. AI/ML Techniques used in infant system

Please view Section 4 Rationale, page 5.

Comment 5:

In all tables, I suggest the authors add information whether the datasets are provided or not, so other researchers can access the datasets they need (instead of only stating some of them in Table 4).

Response:

In most of the studies that are included in Table 2 and 4, are based on the sensor devices that are used to monitor the pregnant women and infants respectively. The last column of these tables basically presents how the data acquired for the study. Mostly, the data is collected from volunteers through sensing devices, which is not available publicly. However, we have updated the tables to make it clear for the readers.

Please view Tables 2 and 4

In Tables 3 and 5, the AI/ML techniques surely use datasets for training and testing the models in the listed studies; however, in most of the studies the authors have used the confidential datasets acquired from some hospital or medical centers, which are not provided for the other researchers. Those studies that used publicly available datasets are mentioned in the Table 6 and corresponding source link is also provided.   

Please view Tables 3, 5 and 6

Comment 6:

Section 4 is the most important part of the paper. The authors must focus on the findings for maternal infant cases, specifically, considering that there are already many studies on sensors and AI/ML for general healthcare. The following questions must be answered well: which findings are specifically new for the maternal infant cases? Sections 4.2 and 4.4 are still highly related to general healthcare and I believe could be found in many other general healthcare papers.

Response:

We are thankful for the valuable comment. Since, in revised version we have added two more sections, please now consider section 4 as section 6. The subheadings of Section 6 were very generic, in the revised version of manuscript, the subheadings of section 6 are updated for clarity. In addition, the content of these subsections is also revised and updated in the context of maternal and infant patients. The section clearly focuses the challenges and opportunities of the review paper.  

Please view Section 6, Page 24.

Comment 7:

When stating the analysis points in Section 4, the authors must always relate to the findings they had in previous sections.

Response:

We are thankful for your comment and agree with it. In fact, this issue appeared due to inadequate flow of the content in the sections 6.1-6.4. (Numbered as 4.1-4.4 in last version of the paper). We have improved it accordingly. It will help readers to relate the findings to the previous sections.

Please see Section 6, Page 24.   

 Comment 8:

The citation numbering must be written in an ascending order. In the manuscript, the authors jump from [16] to [21].

Response:

It was an error in citation insertion, now all the missing citations [17-20] are included.

Please see. Section 1, Page 3, line 75

Comment 9:

Instead of writing "..............." in the tables, please directly write its intended meaning, e.g., "not available", etc.

Response:

In all of the tables “……………” is replaced by not available.

Round 2

Reviewer 2 Report

No further comments. 

Reviewer 3 Report

Thank you fo your revisions.